# Influence of CaF_2_@Al_2_O_3_ on Cutting Performance and Wear Mechanism of Al_2_O_3_/Ti(C,N)/CaF_2_@Al_2_O_3_ Self-Lubricating Ceramic Tools in Turning

**DOI:** 10.3390/ma13132922

**Published:** 2020-06-29

**Authors:** Shuai Zhang, Guangchun Xiao, Zhaoqiang Chen, Chonghai Xu, Mingdong Yi, Qi Li, Jingjie Zhang

**Affiliations:** 1School of Mechanical and Automotive Engineering, Qilu University of Technology (Shandong Academy of Sciences), Jinan 250353, China; 17862989560@163.com (S.Z.); xch@qlu.edu.cn (C.X.); new-raul@163.com (M.Y.); lq1597604438@163.com (Q.L.); zjj@qlu.edu.cn (J.Z.); 2Key Laboratory of Advanced Manufacturing and Measurement and Control Technology for Light Industry in Universities of Shandong, Qilu University of Technology (Shandong Academy of Sciences), Jinan 250353, China

**Keywords:** ceramic tool, coating, wear, cutting performance

## Abstract

This study aimed at improving the cutting performance of a ceramic tool to which were added solid lubricant particles. We prepared the self-lubricating ceramic tool by adding CaF_2_@Al_2_O_3_ instead of CaF_2_, and the self-lubricating ceramic tool with Al_2_O_3_ as matrix phase, Ti(C,N) as reinforcement phase. The properties of the ceramic tool with different contents of CaF_2_@Al_2_O_3_ and CaF_2_ were studied by turning 40Cr. Compared with the ceramic tool with 10 vol.% CaF_2_, the main cutting force and the cutting temperature of the ceramic tool with 10 vol.% CaF_2_@Al_2_O_3_ decreased by 67.25% and 38.14% respectively. The wear resistance and machining surface quality of the ceramic tool with CaF_2_@Al_2_O_3_ were better than the ceramic tool to which were directly added CaF_2_. The optimal content of CaF_2_@Al_2_O_3_ particles was determined to be 10 vol.%. The addition of CaF_2_@Al_2_O_3_ particles effectively reduces the adverse effect of direct addition of CaF_2_ particles on the ceramic tool, and plays a role in improving the cutting performance of the ceramic tool.

## 1. Introduction

With the continuous development of hardened steel in dies, bearings, molds and other industrial fields, researchers have begun to pay attention to the machining of hardened steel. G. K. Dosbaeva et al. [1] compared the performance of WC (tungsten carbide) tools with CVD (chemical vapor deposition) coating and PCBN (polycrystalline cubic boron nitride) tools when turning D2 tool steel. Khaider Bouacha et al. [2] used a CBN (cubic boron nitride) tool to carry out cutting test on AISI (American Iron and Steel Institute) 52,100 bearing steel, and studied the influence of tool wear and cutting parameters on surface roughness. Rachid et al. [3] conducted cutting tests on PM (powder metallurgy) high-speed steel and studied the cutting parameters and conditions suitable for PM high-speed steel. Compared with the traditional grinding method, turning had the characteristics of low cost and high precision [4]. Al_2_O_3_-based ceramic tools had high hardness and good wear resistance, and they were often used to machining hardened steel materials. X. Tian [5] and J. Zhao [6] used Al_2_O_3_-based ceramic tools to turn Inconel 718 and AISI 1045 steel, respectively. If no lubricating fluid was used in the turning process, this avoided the environmental pollution caused by the lubricating fluid [7]. However, in the absence of lubricating fluid, the wear of the ceramic tools was accelerated [8]. As a solution, self-lubrication of ceramic tools can be realized by micro-texture design, coating design and the addition of solid lubricant particles into matrix materials. Kishor Kumar Gajrani et al. [9] filled MoSi_2_ into the mechanical texture of the tool surface to realize the self-lubricating function of the tool. Compared with the tool without MoSi_2_, the tool filled with MoSi_2_ had better cutting performance and lower energy consumption. Xing et al. [10] developed a new cutting tool with WS_2_/Zr coating. The cutting experiment shows that the existence of WS_2_/Zr coating makes the tool self-lubricating and reduces the cutting force in the cutting process. Deng et al. [11] added CaF_2_ to the Al_2_O_3_/TiC ceramic tool matrix to prepare the ceramic tool with lubricating properties, and found that cutting speed had a far-reaching influence on self-lubricating properties.

Adding solid lubricant to the ceramic matrix materials to prepared self-lubricating ceramic tools means that the solid lubricant was directly added to the ceramic matrix materials as a phase, and the multiphase ceramic tools were prepared by sintering methods such as hot-pressing sintering. In machining, solid lubricant was precipitated from the rake face and the flank face of ceramic tools and gradually formed a layer of lubricating film under the extrusion action of the ceramic tool and the workpiece. In ceramic tool materials, CaF_2_ and h-BN have often been added to the ceramic tool matrix materials as lubricating phases. The addition of CaF_2_ and h-BN gave ceramic tools the function of self-lubrication. However, in the multiphase ceramic tool material system, CaF_2_ and h-BN belong to weak phases. Adding CaF_2_ and h-BN to self-lubricating ceramic tools reduced the overall mechanical properties of self-lubricating ceramic tools, which led to the reduction of cutting performance and tool life [12,13]. 

Recently, the application of particle surface modification technology in ceramic materials has provided new insights into solving the problem that solid lubricant (such as CaF_2_ and h-BN) added to ceramic tools reduced the mechanical properties of ceramic tools [14,15,16]. Wu et al. [17,18] coated a layer of Ni on the surface of CaF_2_ particles and h-BN particles, respectively. Tests certified that the microstructure and wear resistance of the ceramic tool material with h-BN@Ni and CaF_2_@Ni were better than those of ceramic matrix to which were directly added h-BN and CaF_2_. Sheng et al. [19] prepared Ni-B@CaF_2_ particles by ultrasonic electrolysis plating. Compared with directly added CaF_2_, the ceramic tool with Ni-B@CaF_2_ particles had better wear resistance. Chen et al. [20,21] prepared h-BN@SiC particles, and prepared the Al_2_O_3_/TiC/h-BN@SiC ceramic tool. The research found that the SiC shell prevented the contact between h-BN and Al_2_O_3_/TiC ceramic matrix, and the bond between the h-BN and Al_2_O_3_/TiC ceramic matrix was closer. Its microstructure and mechanical properties were better than the ceramic tool material with h-BN added. To sum up, the solid lubricant particles can be endowed with new physical and chemical properties by particle surface modification technology. Adding the solid lubricant with core-shell structure to the multiphase ceramic matrix materials can reduce the adverse effect of directly adding solid lubricant to ceramic tools. 

In this paper, the ceramic tool was prepared by adding CaF_2_@Al_2_O_3_ with a core-shell structure to Al_2_O_3_/Ti(C,N) ceramic matrix material. Through a cutting test of 40Cr, the influence of CaF_2_@Al_2_O_3_ addition on the cutting performance of the ceramic tool was studied. The results show that the ceramic tool with CaF_2_@Al_2_O_3_ has better cutting performance than the ceramic tool with CaF_2_. At the same time, we have studied the antifriction and wear resistance mechanism of the ceramic tool with CaF_2_@Al_2_O_3_ in the cutting process. The addition of CaF_2_@Al_2_O_3_ can not only obtain a good friction reduction effect through the precipitation of CaF_2_, but also obtain good wear resistance by coating Al_2_O_3_ on the surface of CaF_2_.

## 2. Materials and Methods 

### 2.1. Preparation of Ceramic Tools

The main materials for preparing ceramic tools and their particle size, purity and manufacturers are listed in Table 1.

The ceramic tool with 5 vol.% CaF_2_@Al_2_O_3_, 10 vol.% CaF_2_@Al_2_O_3_ and 15 vol.% CaF_2_@Al_2_O_3_ were prepared by HP (hot pressing) (sintering temperature 1650 °C, heating rate 20 °C/min, holding time 20 min, hot-pressing pressure 30 MPa). In addition, the ceramic tool with 10 vol.% CaF_2_ was prepared under the same conditions. Table 2 lists the component ratio and mechanical properties of the four ceramic tools.

### 2.2. Cutting Test

The geometric dimensions of the ceramic tools used in this study were 12 mm × 12 mm × 5 mm. Table 3 lists the geometric parameters of the ceramic tools. In this study, the 40Cr hardened steel was selected as the workpiece material for the cutting test. Its hardness was 48–50 HRC, and its main chemical composition is shown in Table 4.

–The cutting tests of 40Cr were carried out with the ATCN-C10, ATCN-C@5, ATCN-C@10 and ATCN-C@15 ceramic tool, respectively. All cutting tests were carried out within the following cutting parameters: depth of cut $\alpha_{p}$ = 0.2 mm, feed rates f = 0.102 mm/r, cutting speed υ = 100 m/min and 300 m/min. The model of the machine tool used was CDE6140A. The Kistler 9265A dynamometer was used to measure the cutting force, and the average value of the cutting force in a stable cutting stage was taken for comparison. The cutting temperature was measured by TH5104 infrared thermal imager. The emission factor was 0.3, and the infrared thermal imager distance from the workpiece was 1.2 m. When the cutting distance reaches 500 m, the maximum temperature was taken for comparison. The tool failure standard was flank wear VB (Wear amount of main flank) = 0.3 mm. TR200 surface roughness measuring instrument was used to measure the workpiece surface roughness Ra. The wear morphology of the ceramic tool was observed by SUPRATM 55 scanning electron microscope, and the elements on the chip were analyzed by energy spectrum analysis.

## 3. Results and Discussion

### 3.1. Microstructure of Ceramic Tool

Figure 1 shows a scanning electron microscope photograph of the fracture surfaces of four ceramic tools, in which the solid lubricant CaF_2_ can be found to agglomerate in the ATCN-C10 ceramic tool with 10 vol.% CaF_2_ added and the ATCN-C@15 ceramic tool with 15 vol.% CaF_2_@Al_2_O_3_ (red squares in Figure 1a,d), which was not conducive to the mechanical properties of the ceramic tool, so the mechanical properties of these two ceramic tools were poor (Table 1). In the fracture surface of the ATCN-C@10 ceramic tool, there are multiple steps (such as 1.2.3.4 point in Figure 1c), which was a typical feature of transgranular fracture. Therefore, the ATCN-C@10 ceramic tool with 10 vol.% CaF_2_@Al_2_O_3_ had both a transgranular fracture and intergranular fracture. Compared with the intergranular fracture of the ATCN-C@5 ceramic tool with 5 vol.% CaF_2_@Al_2_O_3_, this mixed fracture mode is beneficial to improve the mechanical properties of ceramic tools.

Figure 2 is an X-ray diffraction (XRD) pattern of the ATCN-C@10 ceramic tool material. It can be found that the diffraction peaks of Al_2_O_3_ and Ti(C,N) were very obvious, and the diffraction peaks of CaF_2_ can also be clearly observed. The added amount of sintering aid MgO was too small, so no diffraction peak was found. This shows that in the sintering process, there was no obvious chemical reaction between the components of the ceramic cutting tool material, and the components had good chemical compatibility.

### 3.2. Influence of CaF_2_@Al_2_O_3_ on Cutting Force

Figure 3 shows the average numerical comparison of three cutting forces of four ceramic tools when the cutting forces are stable. The axial force (F_x_), the radial force (F_y_) and the main cutting force (F_z_) of the ceramic tool with 10 vol.% CaF_2_@Al_2_O_3_ (ATCN-C@10) were 13.79%, 67.25% and 55.26% lower than the ceramic tool with 10 vol.% CaF_2_ (ATCN-C10), respectively. With the increase of CaF_2_@Al_2_O_3_ content from 5 vol.% to 15 vol.%, the cutting force (F_x_, F_y_, F_z_) tends to decrease first and then increase. When CaF_2_@Al_2_O_3_ content was 10 vol.%, the cutting force (F_x_, F_y_, F_z_) of the ceramic tool was the lowest. This shows that adding an appropriate content of CaF_2_@Al_2_O_3_ was beneficial to reduce the cutting force. Cutting force directly affects ceramic tool wear and surface quality. Compared with the other three ceramic tools, the ATCN-C@10 ceramic tool had lower cutting force, which was conducive to reducing the vibration generated by the ATCN-C@10 ceramic tool in the cutting process and improve the machining quality.

According to the main cutting force *F*z and axial force *F*x measured in the cutting process, the friction coefficient of the tool rake face can be calculated, as shown in Formula (1):(1)$$\mu =\tan\left(\gamma_{0}+\arctan\frac{Fz}{Fx}\right)$$

In the formula, μ is the friction coefficient of the tool rake face, and $\gamma_{0}$ is the rake angle of the tool. It can be seen from Figure 3b that the friction coefficient of the ATCN-C@10 ceramic tool with 10 vol.% CaF_2_@Al_2_O_3_ added was lower than the ATCN-C10 ceramic tool with 10 vol.% CaF_2_ added. By comparing the friction coefficients of the ATCN-C@5, ATCN-C@10 and ATCN-C@15 ceramic tool, it was found that the friction coefficient of the ATCN-C@10 ceramic tool was the lowest, which indicated that the content of solid lubricant was not the main factor affecting the friction coefficient in this ceramic tool system. Adding a proper content of CaF_2_@Al_2_O_3_ can reduce the friction coefficient of ceramic tool rake face.

### 3.3. Influence of CaF_2_@Al_2_O_3_ on Cutting Temperature

At a stable cutting distance of 500 m, the highest temperatures at the tip of four ceramic tools were measured. In Figure 4, it was found that the cutting temperature of the ceramic tool with 10 vol.% CaF_2_@Al_2_O_3_ (ATCN-C@10) added was 38.14% lower than the ceramic tool with 10 vol.% CaF_2_ (ATCN-C10). At higher cutting temperature, 40Cr workpiece material was welded to the ATCN-C10 ceramic tool surface, caused serious adhesive wear.

Figure 5 shows a comparison of the highest temperatures at the tip of four ceramic tools. The cutting temperatures of the three ceramic tools with different contents of CaF_2_@Al_2_O_3_ added were lower than the ceramic tool with 10 vol.% CaF_2_. Added CaF_2_@Al_2_O_3_ can reduced the cutting temperature of ceramic tool. Similar to the change trend of cutting force above, as the content of CaF_2_@Al_2_O_3_ increases from 5 vol.% to 15 vol.%, the cutting temperature tends to decrease and then increase. When the content of CaF_2_@Al_2_O_3_ was 10 vol.%, the cutting temperature was the lowest. This also shows that if an excessive amount of CaF_2_@Al_2_O_3_ was added, there was no benefit to the reduction of cutting temperature. Among the four kinds of ceramic tools, the ceramic tool with 10% CaF_2_@Al_2_O_3_ (ATCN-C@10) added had the lowest cutting temperature, which accelerated the separation speed of chips and rake face, and reduced the wear.

### 3.4. Influence of CaF_2_@Al_2_O_3_ on Flank Wear

As shown in Figure 6a, at a lower cutting speed (υ = 100 m/min), the ceramic tool with 10 vol.% CaF_2_@Al_2_O_3_ (ATCN-C@10) added had the smallest flank wear VB and the longest effective cutting distance. When the cutting distance reached 4500m, the flank wear VB of the ceramic tool with 10 vol.% CaF_2_@Al_2_O_3_ (ATCN-C@10) added was only 0.16 mm while the flank wear VB of the ceramic tool with 15 vol.% CaF_2_@Al_2_O_3_ (ATCN-C@15) was the largest, reaching 0.34 mm, which reached the tool failure standard (VB = 0.30 mm). This was mainly due to the excessive addition of solid lubricant CaF_2_ and the existence of an Al_2_O_3_ shell makes it difficult for CaF_2_ in a ceramic tool to precipitate at a lower cutting speed. As shown in Figure 6b, when the cutting speed increased to 300 m/min, the effect of CaF_2_@Al_2_O_3_ on the flank wear VB increased. The flank wear VB of the ceramic tool with CaF_2_@Al_2_O_3_ added was less than the ceramic tool with CaF_2_ (ATCN-C10), and the slope of wear curve was smaller. This was mainly because the effect of the Al_2_O_3_ shell hindered the precipitation of CaF_2_ weakening under high cutting speed, and the CaF_2_ can be better precipitated and formed a lubricating film on the surface of ceramic tool [22]. The flank wear VB of the ceramic tool with 10 vol.% CaF_2_@Al_2_O_3_ (ATCN-C@10) added was still the smallest.

For the ceramic tool with 10 vol.% CaF_2_ (ATCN-C10) added, the addition of CaF_2_ resulted in a significant reduction in fracture toughness and hardness of the ceramic tool (Table 2), which reduces wear resistance. The high flank wear VB of the ceramic tool with 15 vol.% CaF_2_@Al_2_O_3_ (ATCN-C@15) added was mainly due to the excessive addition of CaF_2_. Although CaF_2_ was coated with a layer of Al_2_O_3_ shell, its mechanical properties were still very low. The difference in flank wear between the ceramic tool with 5 vol.% CaF_2_@Al_2_O_3_ (ATCN-C@5) added and the ceramic tool with 10 vol.% CaF_2_@Al_2_O_3_ (ATCN-C@10) added was mainly due to the small amount of CaF_2_ added in the ATCN-C@5 ceramic tool, which cannot form a compact and uniform lubricating film to reduce tool wear.

### 3.5. Influence of CaF_2_@Al_2_O_3_ on Surface Roughness

Figure 7a,b show the variation of the average value Ra of surface roughness with the cutting distance S in 40Cr turning. On the whole, the surface roughness Ra of four ceramic tools does not exceed 3 μm, which indicated that the addition of CaF_2_ can significantly improve the surface quality. Through previous studies [23], cutting speed was considered to be an important factor affecting surface roughness. As can be seen from Figure 7a, under a low speed cutting condition (υ = 100 m/min), the ceramic tool with 5 vol.% CaF_2_@Al_2_O_3_ (ATCN-C@5) added had the best machining quality, and its surface roughness was kept below 1 μm within 3000 m. Among the four ceramic tools, the ceramic tool with the highest Ra value was the ceramic tool with 15 vol.% CaF_2_@Al_2_O_3_ (ATCN-C@15) added, which was mainly due to its highest flank wear VB (Figure 6a). When the cutting speed was increased from a lower cutting speed (υ = 100 m/min) to a higher cutting speed (υ = 300 m/min), on the one hand, the increase of cutting temperature led to the softening of workpiece materials. On the other hand, CaF_2_ was easier to separate out from ceramic tools. In Figure 7b, the surface roughness Ra of four ceramic tools in the range of 0–3000 m was kept between 0.5 and 1.5 μm. Moreover, the ceramic tool with CaF_2_@Al_2_O_3_ added was better than the ceramic tool with CaF_2_. It shows that the CaF_2_@Al_2_O_3_ can better improve the surface quality at higher cutting speed.

### 3.6. Influence of CaF_2_@Al_2_O_3_ on Wear Morphology and Antifriction Mechanism

The morphology of the rake face of the ceramic tool with 10 vol.% CaF_2_ (ATCN-C10) added and the ceramic tool with 10 vol.% CaF_2_@Al_2_O_3_ (ATCN-C@10) added are presented in the Figure 8a,c. Due to the severe friction between the tool and the chip during the cutting process, the combined action of cutting heat and cutting force resulted in crater wear on the rake face of the two ceramic tools. The ceramic tool with 10 vol.% CaF_2_@Al_2_O_3_ (ATCN-C@10) added had a lower cutting force and lower cutting temperature (Figure 3; Figure 5), so the ceramic tool with 10 vol.% CaF_2_@Al_2_O_3_ (ATCN-C@10) added had a smaller area of crater wear than the ceramic tool with 10 vol.% CaF_2_ (ATCN-C10). There are still regular friction marks on the edge of the stripping area of the ceramic tool with 10 vol.% CaF_2_@Al_2_O_3_ (ATCN-C@10) added, indicating that it can still work after the wear, which does not directly lead to tool failure. At the same time, the rake face of the ceramic tool with 10 vol.% CaF_2_ (ATCN-C10) added also had adhesive wear. This was mainly because the ATCN-C10 ceramic tool had higher cutting temperature and lower fracture toughness, which caused workpiece materials to easily adhere and flake off on the ceramic tool surface.

Figure 8b shows the morphology of the flank face of the ceramic tool with 10 vol.% CaF_2_ (ATCN-C10) added. There was a micro-chipping in the red circle part of Figure 7b, which may be due to the ceramic tool encountered a hard point and suffered great impact. A scribed groove was formed in the flank face near the tip and the cutting edge, which was a typical feature of abrasive wear. As an indenter, the abrasive grains generate cracks on the tool surface, and then the tool wears due to the development and intersection of cracks. At the position away from the cutting edge and the tip, the ceramic tool with 10 vol.% CaF_2_ (ATCN-C10) added experienced adhesive wear. From Figure 8d, it was found that the flank wear of the ceramic tool with 10 vol.% CaF_2_@Al_2_O_3_ (ATCN-C@10) added was relatively light. There was only slight abrasive wear and crater wear. The overall wear of the ceramic tool with 10 vol.% CaF_2_@Al_2_O_3_ (ATCN-C@10) added was better than the ceramic tool with 10 vol.% CaF_2_ (ATCN-C10), and it shows better wear resistance.

Figure 9a shows the morphology of the chips when machining 40Cr steel with the ceramic tool with 10 vol.% CaF_2_@Al_2_O_3_ (ATCN-C@10) added. It can be seen that the chips were slightly bent, mainly strip-shaped chips, but the edges of the chips were obviously jagged. The cutting process of forming strip chips was relatively smooth, the fluctuation of the cutting force was small, and the quality of the machined surface was better. In Figure 9b, the content of O element was relatively high, which indicated that the chip surface was oxidized. According to the research of Deng et al. [9], the distribution of the F element can prove that CaF_2_ precipitates on the tool surface. In Figure 9b, it can be seen that the F element precipitates well on the tool surface, which indicated that the precipitation of CaF_2_ in the ceramic tool was good.

Figure 10 shows the antifriction and wear resistance mechanism of the self-lubricating ceramic tool. In the initial stage of cutting, the CaF_2_@Al_2_O_3_ particles uniformly distributed in the ceramic matrix will not precipitate out of the tool to form a lubricating film (Figure 10a). Therefore, at the cutting distance of 0–200 m, the flank wear VB increased rapidly (Figure 6). Then, under the action of cutting force, as shown in Figure 10b, the Al_2_O_3_ shell of CaF_2_@Al_2_O_3_ particles was destroyed, and CaF_2_ was exposed on the surface of the ceramic tool. The CaF_2_ starts to precipitate from the surface of the ceramic tool in Figure 10c. However, at this time, only a small amount of CaF_2_ was precipitated and discontinuous and incomplete lubricating film was formed. With the continuous progress of the cutting process, the cutting temperature keeps rising, which makes CaF_2_ change from brittle state to plastic state, forming a lubricating film with CaF_2_ as the main body on the surface of the tool (Figure 10d). The elemental analysis of the chips in Figure 9b also confirmed the presence of CaF_2_ on the tool surface. Due to the low shear strength of the lubricating film, it had good antifriction effect during cutting. Therefore, after the cutting distance reached 200 m, the flank wear rate became slower (Figure 6).

## 4. Conclusions

This paper studied the influence of adding CaF_2_@Al_2_O_3_ on cutting performance of ceramic tool by turning 40Cr. The surface morphology and chip morphology of the ceramic tool were observed and analyzed. The following conclusions are drawn from the above research results.

(1) The cutting force, cutting temperature, surface roughness and flank wear of ceramic tools with CaF_2_@Al_2_O_3_ added were lower. Therefore, adding CaF_2_@Al_2_O_3_ to the ceramic tool was more beneficial to improve the cutting performance of the ceramic tool than directly adding CaF_2_ to the ceramic tool. At higher cutting speed, the effect of the Al_2_O_3_ shell on preventing CaF_2_ precipitation will be weakened. The ceramic tool with CaF_2_@Al_2_O_3_ added had better cutting performance at higher cutting speeds.

(2) As the content of CaF_2_@Al_2_O_3_ increased from 5 vol.% to 15 vol.%, the cutting force and cutting temperature of the ceramic tools showed a downward trend and increased. When the content of CaF_2_@Al_2_O_3_ is 10vol.%, the cutting force and cutting temperature of ceramic tools were the lowest, and it had better wear resistance and processed surface quality. The ceramic tool with 10 vol.% CaF_2_@Al_2_O_3_ added mainly suffers from crater wear and adhesive wear, and the chip morphology was strip chip.

(3) By coating a layer of Al_2_O_3_ on the surface of CaF_2_ particles, the hardness, flexural strength and fracture toughness of the ceramic tool are improved and it has good wear resistance. For the ceramic tool with CaF_2_@Al_2_O_3_ added, as the cutting process progresses, the Al_2_O_3_ shell in CaF_2_@Al_2_O_3_ breaks and CaF_2_ precipitates from the tool, forming a lubricating film on the tool surface, which gives the ceramic tool a good anti-friction effect.

## Figures and Tables

**Figure 1 materials-13-02922-f001:**
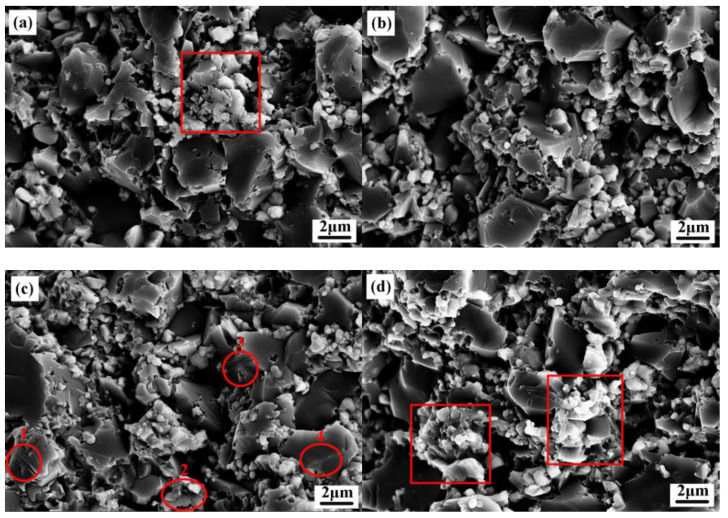
(**a**–**d**) Scanning electron microscope (SEM) photograph of fracture surfaces of four kinds of four ceramic tools.

**Figure 2 materials-13-02922-f002:**
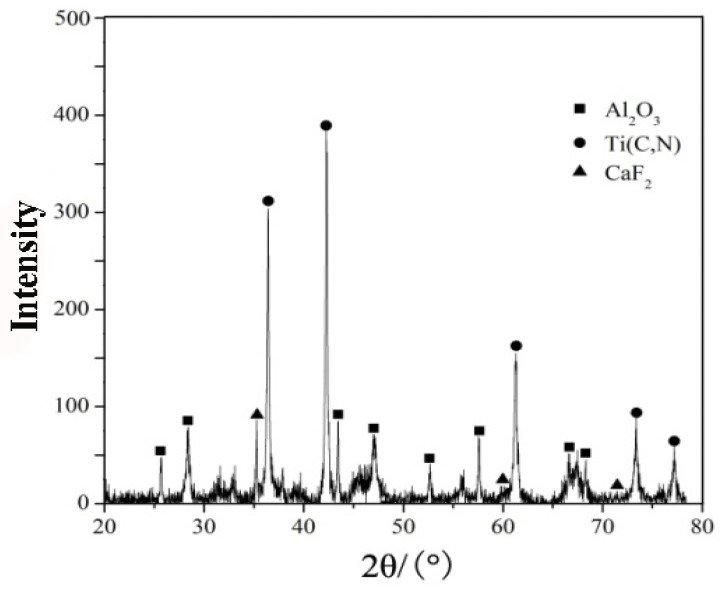
X-ray diffraction (XRD) pattern of the ATCN-C@10 ceramic tool with 10 vol.% CaF_2_@Al_2_O_3_ added_._

**Figure 3 materials-13-02922-f003:**
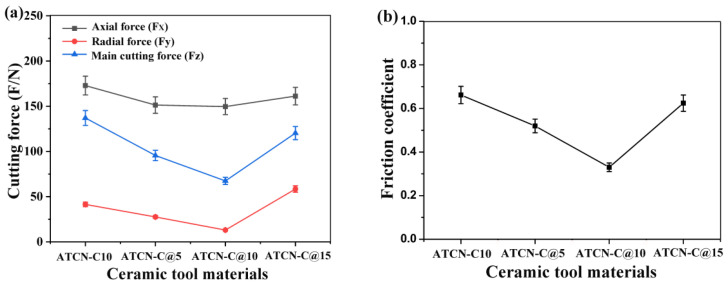
Comparison of (**a**) cutting forces and (**b**) friction coefficient of four ceramic cutting tools (test conditions: depth of cut $\alpha_{p}$ = 0.2 mm, feed rates f = 0.102 mm/r, cutting speed υ = 300 m/min).

**Figure 4 materials-13-02922-f004:**
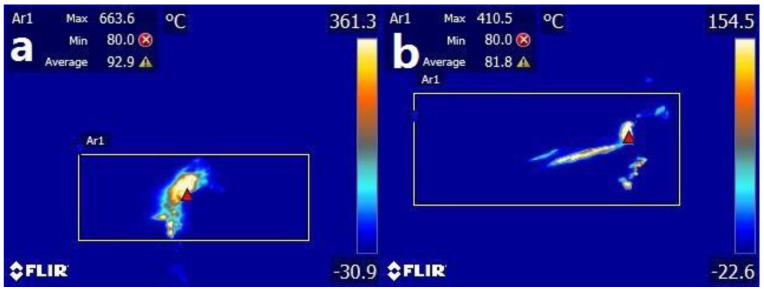
Comparison of cutting temperatures between (a) ATCN-C10 and (b) ATCN-C@10 ceramic tools (test conditions: depth of cut $\alpha_{p}$ = 0.2 mm, feed rates f = 0.102 mm/r, cutting speed υ = 300 m/min).

**Figure 5 materials-13-02922-f005:**
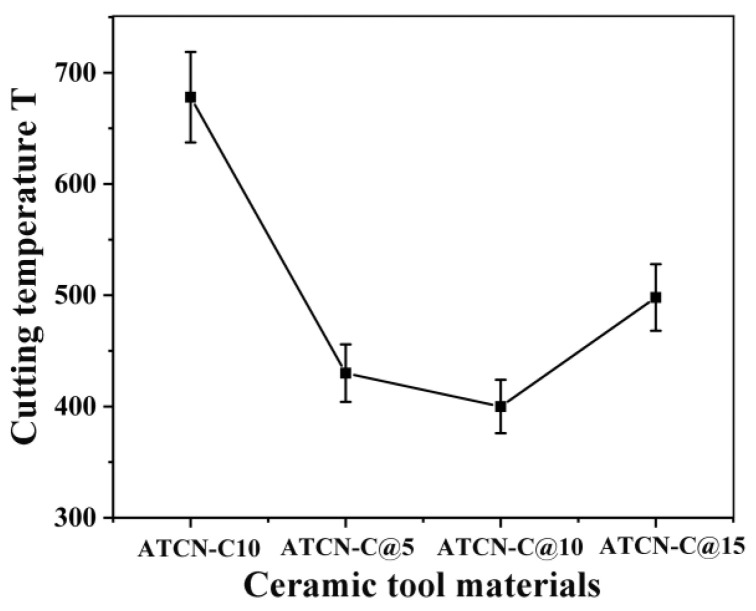
Comparison of cutting forces of four ceramic cutting tools (test conditions: depth of cut $\alpha_{p}$ = 0.2 mm, feed rates f = 0.102 mm/r, cutting speed υ = 300 m/min).

**Figure 6 materials-13-02922-f006:**
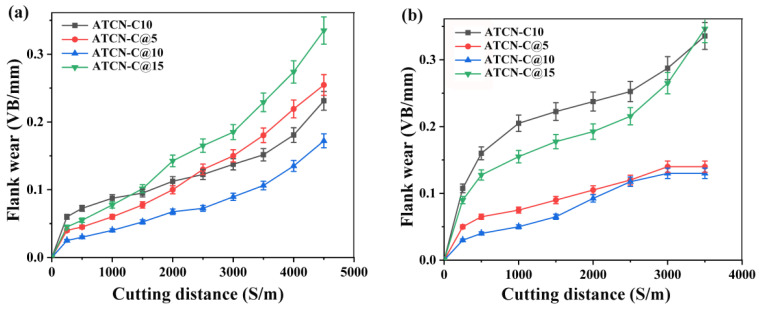
The flank wear of four ceramic tools at cutting speeds of (**a**) 100 m/min and (**b**) 300 m/min (test conditions: depth of cut $\alpha_{p}$ = 0.2 mm, feed rates f = 0.102 mm/r).

**Figure 7 materials-13-02922-f007:**
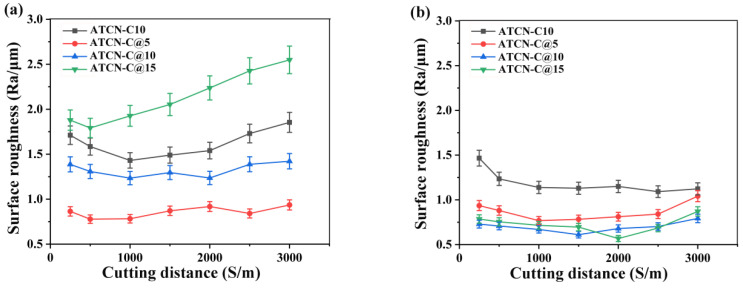
The surface roughness of four ceramic tools at cutting speeds of (**a**) 100 m/min and (**b**) 300 m/min (test conditions: depth of cut $\alpha_{p}$ = 0.2 mm, feed rates f = 0.102 mm/r).

**Figure 8 materials-13-02922-f008:**
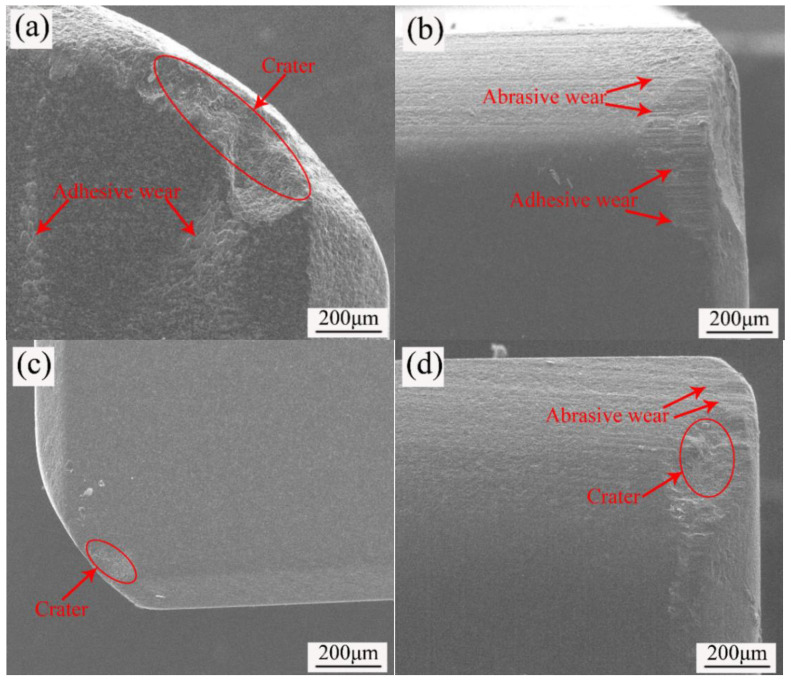
Wear profile of the rake faces of (**a**) the ATCN-C10 ceramic tool and (**c**) the ATCN-C@10 ceramic tool, the flank faces of (**b**) the ATCN-C10 ceramic tool and (**d**) the ATCN-C@10 ceramic tool (test conditions: depth of cut $\alpha_{p}$ = 0.2 mm, feed rates f = 0.102 mm/r, υ = 300 m/min).

**Figure 9 materials-13-02922-f009:**
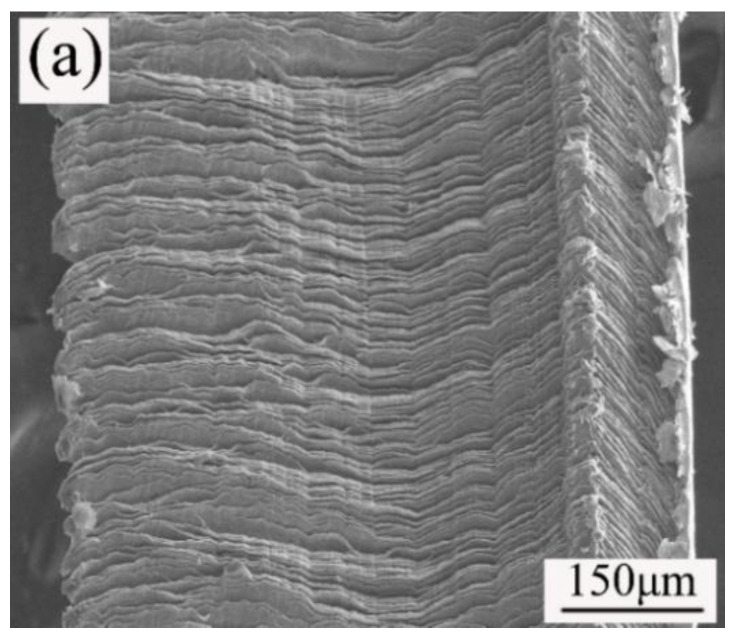
(**a**) Chip morphology and (**b**) energy spectrum analysis of the ATCN-C@10 ceramic tool (test conditions: depth of cut = 0.2 mm, feed rates f = 0.102 mm/r, υ = 300 m/min).

**Figure 10 materials-13-02922-f010:**
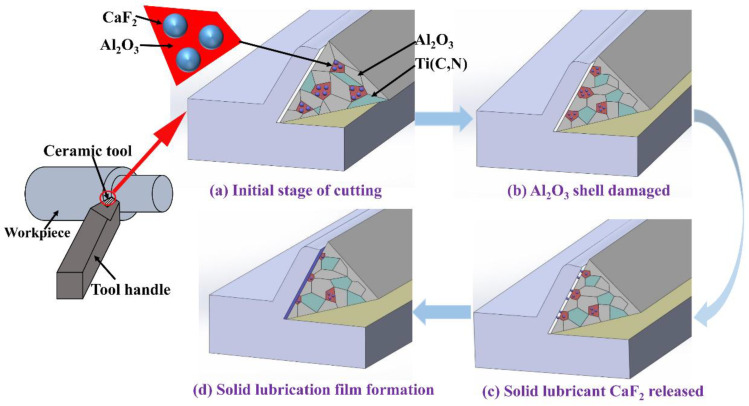
Schematic diagram of solid lubricating film formation process of the ceramic tool with CaF_2_@Al_2_O_3_ added: (**a**) initial stage of cutting, (**b**)Al_2_O_3_ shell damaged, (**c**) solid lubricant CaF_2_ released and (**d**) solid lubricating film formation.

**Table 1 materials-13-02922-t001:** Main materials for preparing ceramic tool materials.

Name	Particle Size	Purity	Manufacturer
Al_2_O_3_	1 µm	>99.9%	Shanghai Chaowei Nanotechnology Co., Ltd.(Shanghai, China)
Ti(C,N)	0.5 µm	>99.9%	Shanghai Chaowei Nanotechnology Co., Ltd.(Shanghai, China)
MgO	1 µm	>99.9%	Chemical Reagents Co., Ltd.(Beijing, China)
CaF_2_@Al_2_O_3_	1-5 µm	-	self-made

**Table 2 materials-13-02922-t002:** Component ratio and mechanical properties of ceramic tools.

Sample	Main Chemical Components (vol.%)					Mechanical Properties		
	Al_2_O_3_	Ti(C,N)	MgO	CaF_2_	CaF_2_@Al_2_O_3_	Hardness/(GPa)	Flexural Strength/(MPa)	Fracture Toughness/(MPa•m^1/2^)
ATCN-C10	67.34	22.16	0.5	10	-	15.10	606	5.02
ATCN-C@5	71.10	23.40	0.5	-	5	16.31	632	6.25
ATCN-C@10	67.34	22.16	0.5	-	10	17.29	680	6.50
ATCN-C@15	63.58	20.92	0.5	-	15	15.53	571	5.62

**Table 3 materials-13-02922-t003:** Ceramic tool geometric parameters.

Rake Angle $\gamma_{0}$	Relief Angle $\alpha_{0}$	Cutting Edge Angle κ_r_	Corner Radius $\gamma_{\varepsilon }$	Inclination Angle λ_S_	Cutting Negative Chamber $b_{\gamma 1}\times \gamma_{01}$
−5°	5°	45°	0.2 mm	0°	0.1 mm × −10°

**Table 4 materials-13-02922-t004:** Composition of workpiece material 40Cr (wt.%).

Workpiece	C	Si	Mn	Cr	Ni	S	P	Fe
40 Cr	0.37–0.45	0.17–0.37	0.5–0.8	0.8–1.1	≤0.03	≤0.035	≤0.035	Bal.

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
