# Peer review of "Influence of CaF2@Al2O3 on Cutting Performance and Wear Mechanism of Al2O3/Ti(C,N)/CaF2@Al2O3 Self-Lubricating Ceramic Tools in Turning"

_materials, 2020, doi:10.3390/ma13132922_

Round 1

Reviewer 1 Report

The manuscript entitled: 'Influence of [email protected] on Cutting Performance and Wear Mechanism of Al2O3/Ti(C,N)/[email protected] Self-lubricating Ceramic Tools in Turning' reports the influence of the addition of self-lubricating ceramic on the tribological properties and its mechanism. I have the following minor concerns with the manuscript.

  • XRD of the material should be introduced to show the presence of the phases in the sample.
  • Error bars should be introduced for all the data points.
  • Typos in the manuscript should be rectified. For instance in Abstract: 10vol% should be written as 10 vol.%.
  • English language needs attention.

Author Response

Point 1: XRD of the material should be introduced to show the presence of the phases in the sample.

Response 1: Thank you for your advice. we have added the XRD pattern of the material in lines 130-137 of the manuscript.

Point 2: Error bars should be introduced for all the data points.

Response 2: As Reviewer suggested, we have added error bars for all data points in the manuscript.

Point 3: Typos in the manuscript should be rectified. For instance in Abstract: 10vol% should be written as 10 vol.%.

Response 3: Thank you for your advice, we have already corrected it in the manuscript.

Point 4: English language needs attention.

Response 4: Thank you for your advice. After careful examination, we have corrected many grammatical errors in the article.

Reviewer 2 Report

The manuscript looks in good shape. However, consider the points for further improvements;

  • Add more details and captions about figure 9. 
  •  Since you have the forces can you add coefficient of friction graphs as well?

Author Response

Point 1: Add more details and captions about figure 9.

Response 1: Thank you for your advice, we have added more details and comments to Figure 9, as follows:

Figure 10 shows the antifriction and wear resistance mechanism of the self-lubricating ceramic tool. In the initial stage of cutting, the CaF2@Al2O3 particles uniformly distributed in the ceramic matrix will not precipitate out of the tool to form a lubricating film (Figure 10(a)). Therefore, at the cutting distance of 0-200m, the flank wear VB increased rapidly (Figure 6). Then, under the action of cutting force, as shown in Figure 10 (b), the Al2O3 shell of CaF2@Al2O3 particles is destroyed, and CaF2 is exposed on the surface of the ceramic tool. The CaF2 starts to precipitate from the surface of the ceramic tool in Figure 10(c). However, at this time, only a small amount of CaF2 was precipitated and discontinuous and incomplete lubricating film was formed. With the continuous progress of the cutting process, the cutting temperature keeps rising, which makes CaF2 change from brittle state to plastic state, forming a lubricating film with CaF2 as the main body on the surface of the tool (Figure 10(d)). The elemental analysis of the chips in Figure 9(b) also confirmed the presence of CaF2 on the tool surface. Due to the low shear strength of the lubricating film, it had good antifriction effect during cutting. Therefore, after the cutting distance reached 200m, the flank wear rate becalmed slower (Figure 6).

Point 2: Since you have the forces can you add coefficient of friction graphs as well?

Response 2: Thank you for your advice, we have added friction coefficient graphs of rake faces of four kinds of ceramic tools.

Reviewer 3 Report

I consider the article interesting and highly up to date and also I see the possibility of transferring knowledge directly into industrial practice. Article has form similar to the report from a research but processing method suits me. Title is adequate to the contribution. The authors have created sufficient theoretical basis for carrying out the mentioned analyzes. The theoretical background is understandable and used literary sources are relevant. However, in my opinion, Introduction section should be improved. Section "Materials and methods" have been described in detail. In the experiments were used the latest measurement technology and methodologies which contributes to the relevance of the data. Article contains necessary figures and tables. I consider the article suitable for publishing in the journal "Materials", but some details are missing:

- In my opinion Introduction should be improved. I suggest add more information and examples to better describe what other researchers have done in this area. References should be done in one to one mode for better clarity (for example line 29, 31, 32, 36). Also, chapter " Results and discussion " could be more combined with other published articles and the results achieved by other authors.

In Materials and Methods

- How the cutting parameters for tests were selected and adopted?

- Was adopted the experimental test plan in the research?

- Are the presented results (in Figures 2,4,5 and 6) mean values? This is also not clear in the text. What was the dispersion of results?

- In 3.3. Influence of [email protected] on cutting temperature - What were the conditions for thermovision measurements of temperature in the cutting zone and what were the values of the measurement parameters? There is no information about the emission factor, distance from the workpiece, etc. Are the temperature values presented in Fig. 4 the maximum values?

- In Conclusions: Authors should give some examples of data to confirm the conclusions.

Author Response

Point 1: In my opinion Introduction should be improved. I suggest add more information and examples to better describe what other researchers have done in this area. References should be done in one to one mode for better clarity (for example line 29, 31, 32, 36). Also, chapter " Results and discussion " could be more combined with other published articles and the results achieved by other authors.

Response 1: Thank you for your advice, we have revised the introduction. Lines 28-84 are now amended to read as follows:  

With the continuous development of hardened steel in dies, bearings, molds and other industrial fields, researchers have begun to pay attention to the machining of hardened steel. G. K. Dosbaeva et al. [1] compared the performance of WC tools with CVD coating and PCBN tools when turning D2 tool steel. Khaider Bouacha et al. [2] used CBN tool to carry out cutting test on AISI 52100 bearing steel, and studied the influence of tool wear and cutting parameters on surface roughness. Rachid et al. [3] conducted cutting tests on PM (powder metallurgy) high speed steel and studied the cutting parameters and conditions suitable for PM high speed steel. Compared with the traditional grinding method, turning had the characteristics of low cost and high precision [4]. Al2O3-based ceramic tools had high hardness and good wear resistance, they were often used to machining hardened steel materials, X. Tian and J. Zhao used Al2O3-based ceramic tools to turning Inconel 718 and AISI 1045 steel respectively [5,6]. If no lubricating fluid was used in the turning process, this avoided the environmental pollution caused by the lubricating fluid [7]. However, in the absence of lubricating fluid, the wear of ceramic tools was accelerated [8]. As a solution, self-lubrication of ceramic tools can be realized by micro-texture design, coating design and added solid lubricant particles into matrix materials. Kishor Kumar Gajrani et al. [9] filled MoSi2 into the mechanical texture of the tool surface to realize the self-lubricating function of the tool. Compared with the tool without MoSi2, the tool filled MoSi2 had better cutting performance and lower energy consumption. Xing et al. [10] developed a new cutting tool with WS2/Zr coating. The cutting experiment shows that the existence of WS2/Zr coating makes the tool self-lubricating and reduces the cutting force in the cutting process. Deng et al. [11] added CaF2 to Al2O3/TiC ceramic tool matrix to prepare the ceramic tool with lubricating properties, and found that cutting speed had far-reaching influence on self-lubricating properties.

Added solid lubricant to the ceramic matrix materials to prepared self-lubricating ceramic tools means that the solid lubricant was directly added to the ceramic matrix materials as a phase, and the multiphase ceramic tools were prepared by sintering methods such as hot pressing sintering. In machining, solid lubricant was precipitated from the rake face and the flank face of ceramic tools and gradually forms a layer of lubricating film under the extrusion action of the ceramic tool and the workpiece. In ceramic tool materials, CaF2 and h-BN were often added to the ceramic tool matrix materials as lubricating phases. The addition of CaF2 and h-BN made ceramic tools have the function of self-lubrication. However, in the multiphase ceramic tool material system, CaF2 and h-BN belong to weak phases. Added CaF2 and h-BN to self-lubricating ceramic tools reduced the overall mechanical properties of self-lubricating ceramic tools, which led to the reduction of cutting performance and tool life [12,13].

Recently, the application of particle surface modification technology in ceramic materials provided new enlightenment for solved the problem that solid lubricant (such as CaF2 and h-BN) was added to ceramic tools reduced the mechanical properties of ceramic tools [14-16]. Wu et al. [17,18] coated a layer of Ni on the surface of CaF2 particles and h-BN particles respectively. Test certificate that the microstructure and wear resistance of the ceramic tool material added with [email protected] and CaF2@Ni were better than those of ceramic matrix directly added with h-BN and CaF2. Sheng et al. [19] prepared [email protected]2 particles by ultrasonic electroless plating. Compared with directly added CaF2, the ceramic tool added with [email protected]2 particles had better wear resistance. Chen et al. [20,21] prepared [email protected] particles, and prepared the Al2O3/TiC/[email protected] ceramic tool. The research found that the SiC shell prevented the contact between h-BN and Al2O3/TiC ceramic matrix, and the bond between h-BN and Al2O3/TiC ceramic matrix was closer. Its microstructure and mechanical properties were better than the ceramic tool material added with h-BN. To sum up, the solid lubricant particles can be endowed with new physical and chemical properties by particle surface modification technology. Added the solid lubricant with core-shell structure to the multiphase ceramic matrix materials can reduced the adverse effect of directly added solid lubricant to ceramic tools.

In this paper, the ceramic tool was prepared by adding CaF2@Al2O3 with core-shell structure to Al2O3/Ti(C,N) ceramic matrix material. Through the cutting test of 40Cr, the influence of CaF2@Al2O3 addition on the cutting performance of ceramic tool was studied. The results show that the ceramic tool with CaF2@Al2O3 has better cutting performance than the ceramic tool with CaF2. At the same time, we have studied the antifriction and wear resistance mechanism of the ceramic tool added with CaF2@Al2O3 in the cutting process. The addition of CaF2@Al2O3 can not only obtain good friction reduction effect through the precipitation of CaF2, but also obtain good wear resistance by coating Al2O3 on the surface of CaF2.

As Reviewer suggested, we have intensified the discussion on the results. First of all, in line 215, we added the prediction of the influence of cutting speed on surface roughness. Secondly, in line 265, I quoted other people's literature to further prove that CaF2 precipitation is good

Point 2: How the cutting parameters for tests were selected and adopted?

Response 2: Thank you for your advice, in section 2.2, we added cutting parameters selected for cutting tests.

Point 3: Was adopted the experimental test plan in the research?

Response 3: Thank you for your advice, the cutting tests of 40Cr were carried out with the ATCN-C10, [email protected], [email protected] and [email protected] ceramic tool, respectively. All cutting tests were carried out within the following cutting parameters: depth of cut = 0.2 mm, feed rates =0.102 mm/r, cutting speed =100m/min and 300 m/min. The cutting force, cutting temperature, flank wear and surface roughness of four kinds of ceramic tools are compared. The representative ATCN-C10 ceramic tool and [email protected] ceramic tool are selected from four kinds of ceramic tools to analyze the tool surface wear morphology. Finally, the [email protected] ceramic tool with the best cutting performance is selected to analyse the antifriction mechanism.

Point 4: Are the presented results (in Figures 2,4,5 and 6) mean values? This is also not clear in the text. What was the dispersion of results?

Response 4: Figure 2 is the average value of the three cutting forces after the cutting forces have stabilized. Figure 4 shows the highest temperature at the tip of the ceramic tool when the cutting distance reaches 500m. Figure 5 is the average value of the tool flank wear. Figure 6 is the average value of the surface roughness. We have added the above information to the manuscript.

Point 5: In 3.3. Influence of CaF2@Al2O3 on cutting temperature - What were the conditions for thermovision measurements of temperature in the cutting zone and what were the values of the measurement parameters? There is no information about the emission factor, distance from the workpiece, etc. Are the temperature values presented in Fig. 4 the maximum values?

Response5Thank you for your advice, we have added a test method for cutting temperature in section 2.2 of the article: The cutting temperature in the cutting process is measured by an infrared thermal imager (model Flir-A320). The emission factor is 0.3, and the infrared thermal imager distance from the workpiece is 1.2m. After the cutting distance of the tool reaches 500 m, the maximum value of the tip temperature is selected to compare the cutting temperatures.

Point 6: In Conclusions: Authors should give some examples of data to confirm the conclusions.

Response6: Thank you for your advice, we added some examples of data in the conclusion Lines 290-310 are now amended to read as follows:

This paper studied the influence of adding CaF2@Al2O3 on cutting performance of ceramic tool by turning 40Cr. The surface morphology and chip morphology of the ceramic tool were observed and analyzed. The following conclusions are drawn from the above research results.

(1) The cutting force, cutting temperature, surface roughness and flank wear of ceramic tools added with CaF2@Al2O3 were lower. Therefore, Added CaF2@Al2O3 to the ceramic tool was more beneficial to improve the cutting performance of the ceramic tool than directly added CaF2 to the ceramic tool. At higher cutting speed, the effect of Al2O3 shell on preventing CaF2 precipitation will be weakened. The ceramic tool added with CaF2@Al2O3 had better cutting performance at higher cutting speeds.

(2) As the content of CaF2@Al2O3 increased from 5 vol.% to 15 vol.%, the cutting force and cutting temperature of ceramic tools showed a downward trend and increased. When the content of CaF2@Al2O3 is 10vol.%, the cutting force and cutting temperature of ceramic tools are the lowest, and it has better wear resistance and processed surface quality. The ceramic tool added with 10 vol.% CaF2@Al2O3 mainly suffers from crater wear and adhesive wear. And, the chip morphology was strip chip.

(3) By coating a layer of Al2O3 on the surface of CaF2 particles, the hardness, flexural strength and fracture toughness of the ceramic tool are improved and it has good wear resistance. For the ceramic tool added with CaF2@Al2O3, as the cutting process progresses, the Al2O3 shell in CaF2@Al2O3 breaks and CaF2 precipitates from the tool, forming a lubricating film on the tool surface, which makes the ceramic tool have a good anti-friction effect.
